# Distinguishing probabilistic from non-probabilistic neural representations

**Ishan Kalburge**
Department of Engineering
University of Cambridge
ik437@cam.ac.uk

**Máté Lengyel**
Department of Engineering     Department of Cognitive Science
University of Cambridge     Central European University
m.lengyel@eng.cam.ac.uk

## Abstract

The precise neural mechanisms of probabilistic computation remain unknown despite growing evidence that humans track their uncertainty. Recent work has proposed that probabilistic representations arise naturally in task-optimized neural networks. However, previous decoding approaches only tested sufficiency—whether posteriors were decodable from neural activity—without testing whether these representations were minimal—whether they filter irrelevant input information. This limitation makes it difficult to distinguish genuine probabilistic representations from trivial input recoding. We introduce the functional information bottleneck (fIB) framework, which evaluates neural representations based on both sufficiency (posterior decodability) and minimality (invariance to irrelevant inputs). Using this novel approach, we show networks trained to perform cue combination, coordinate transformation, and Kalman filtering without probabilistic objectives encode Bayesian posteriors in their hidden layer activities, but these networks fail to compress their inputs in a task-optimal way, instead performing heuristic computations akin to input re-representation. Therefore, it remains an open question under what conditions truly probabilistic representations emerge in neural networks. More generally, our work provides a stringent framework for identifying probabilistic codes, and lays the foundation for systematically examining whether, how, and which posteriors are represented in neural circuits during complex decision-making.

## 1   Introduction

Under a particular generative model of the world that prescribes how latent variables generate observations, the human brain may employ one of two broad classes of recognition models to estimate those latent variables [1]: probabilistic models that employ Bayesian inference [2], and non-probabilistic models that compute intermediate values that do not necessarily correspond to posteriors over latent variables, such as function-approximating neural networks [3]. Although behavioral evidence suggests that human and non-human primates are uncertainty-aware in perceptual judgements [4–6], it remains unclear whether uncertainty is represented probabilistically—*i.e.,* that neural circuits themselves compute with probabilities—or heuristically through dedicated channels for processing uncertainty [7, 8].

Recent work has suggested that neural networks develop robust internal representations of posteriors even without explicit probabilistic inductive biases [9], suggesting that probabilistic representation is an emergent phenomenon of near-optimal behavior. However, previous decoding approaches [9, 10] did not distinguish truly probabilistic representations from trivial re-representations of inputs.

Here, we seek to define what aspect of the neural code distinguishes a probabilistic representation from a non-probabilistic one. Recent debates in probabilistic neural coding have clarified that defining probabilistic representation ultimately depends on how we define representation itself [8, 7, 11].

Preprint.

Building on this view, we link general criteria for representation to an information bottleneck perspective on probabilistic inference. From this perspective, probabilistic neural coding can be meaningfully identified only when neural activity forms approximately minimal sufficient codes—codes that preserve exactly the information needed for behavior and generalization, and nothing more. This reflects a long-standing idea that inference is a form of compression [2], but one that has rarely been applied to distinguish neural representations of uncertainty. Here, we demonstrate how this compression-based view can sharpen our understanding of probabilistic neural coding.

To circumvent challenges with existing information bottleneck analyses [12–14], we leverage linear and nonlinear probing, widely used in machine learning and mechanistic interpretability [15, 16], to assess information content in networks optimized to perform probabilistic tasks. Using our novel approach, we study a variety of task-optimized neural networks that had been suggested to develop probabilistic representations in earlier work [9]: networks trained to perform static inference tasks (such as cue combination and coordinate transformation) or dynamic state estimation tasks (Kalman filtering). While these tasks and their corresponding networks are relatively simple, we chose them because 1) they have tractable and analytic generative models, which are crucial for the validating the fIB framework and 2) they have been studied extensively in neuroscience and are tasks in which human (and other animal) behavior has been shown to be uncertainty-aware in a Bayesian manner [4, 17–19]. Crucially, the minimality criterion inherent in our fIB approach reveals that, contrary to the findings of [9], task-optimized neural networks do not generically form probabilistic representations.

## 2 An information bottleneck approach to probabilistic representation

For a function, $\mathbf{r} = f(\mathbf{X})$, to be a probabilistic representation, where $\mathbf{X}$ denotes the inputs to the system and $p_z = p(z|\mathbf{X})$ is some target posterior over a relevant latent variable $z$, we expect $\mathbf{r}$ to exhibit sufficiency for $\mathbf{y}$ and invariance to nuisances $\mathbf{n}$ [20, 11]. Sufficiency is expressed as $I(\mathbf{r}; p_z) = I(\mathbf{X}; p_z)$, or that the representation $\mathbf{r}$ is maximally informative about $p_z$. Invariance enforces that $\mathbf{r}$ filters out nuisance variables $\boldsymbol{\nu}$, *i.e.,* $I(p_z; \boldsymbol{\nu}) \approx 0$.

Invariance is often tested by holding specific nuisance variables fixed [10] or by designing tasks where optimal performance should be independent of nuisance variation and then evaluating out-of-distribution generalization [9]. While compelling, these approaches make it difficult to exhaustively cover all possible nuisances or distributional shifts. Moreover, they do not address in-distribution redundancy: from a Bayesian perspective, compression arises not only from marginalizing over nuisance variables but also from eliminating redundant information in $\mathbf{X}$ about $\mathbf{y}$. This redundancy reduction ensures that downstream readouts remain robust to distributional changes and that Bayesian modules can transfer flexibly across inference architectures without heavy fine-tuning.

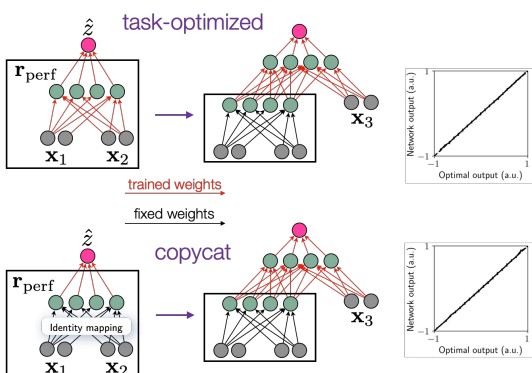

Figure 1: Extension of the task transfer setting from [9]. To assess whether a neural representation is probabilistic, it is necessaryt ot test minimality in order to rule out trivial 'copycat' strategies.

Therefore, a simpler—and stronger—test of invariance is minimality (or compression): a code that is minimal with respect to input information but sufficient with respect to task-relevant inference variables is necessarily invariant to all nuisance variation [20]. Thus, probabilistic representation—sufficiency and invariance— can be viewed as a relaxation of minimal sufficiency similar to an information-bottleneck-style criterion:

$$\arg\max_{\mathbf{r}} I(\mathbf{r}; p_z) \quad \text{subject to} \quad I(\mathbf{X}; \mathbf{r}) \leq \alpha.$$

A central advantage of probabilistic representations is their potential for flexible reuse across tasks. [1] For example, an agent trained to estimate the latent stimulus $z$ driving two cues $\mathbf{X} = (\mathbf{x}_1, \mathbf{x}_2)$ should be able to reuse its internal representation if it needs to perform such cue combination with access to a third cue $\mathbf{x}_3$. This is implicitly a test for whether a representation of $\mathbf{X}$ is *sufficient* for representing $p_z$. Indeed, [9] shows that when a network trained on $\mathbf{X}$ to estimate $z$ is *frozen*

and its hidden representation $\mathbf{r}_{\text{perf}}$ is grafted onto another network with a third cue $\mathbf{x}_3$, this modular network optimally performs three-cue combination (Figure 1A, top). However, the same three-cue network performs equally well if $\mathbf{r}_{\text{perf}}$ simply encodes $\bar{\mathbf{X}}$ itself (Figure 1A, bottom). In other words, $\mathbf{X}$ itself is trivially a sufficient statistics for $p_z$. Thus, only testing whether the task-relevant posterior is decodable from $\mathbf{r}_{\text{perf}}$ is insufficient in assessing probabilistic representation because a sufficiently expressive decoder can trivially decode the optimal posterior if $\mathbf{r}_{\text{perf}}$ encodes $\mathbf{X}$ instead of $p(z|\mathbf{X})$. Instead, a specific probabilistic representation—one that is identifiable and nontrivial—must also compress away spurious correlations in $\mathbf{X}$ that would otherwise interfere with downstream inference and flexible reuse.

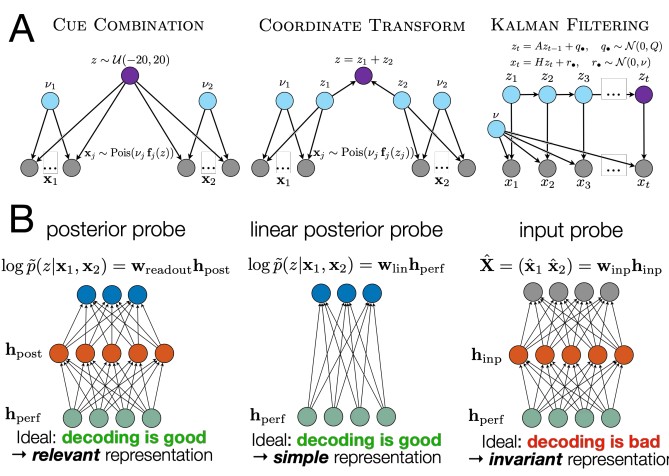

Figure 2: All tasks and probes used in our analysis. Networks were trained to invert the generative models shown in **A)**. **B)** The hidden representations of these networks were probed for minimal sufficiency using our fIB framework.

It is necessary to point out a few key distinctions here between our framework and the classical information bottleneck literature [21–23]. First, we do not assume $\mathbf{r}$ is a stochastic encoding of $\mathbf{X}$ [24]. Two, we do not explicitly train any of our task-optimized networks with an IB objective, as has been proposed in [25], so we do not choose $\alpha$; our analyses are all post-hoc. Third, we do **not** attempt to estimate mutual information directly (i.e., using problematic methods such as those in [26, 12], especially given that mutual information is a vacuous quantity for deterministic neural networks [13, 14]. We instead rely on probing decoders to approximate information content in the hidden layers of task-optimized networks.

## 2.1 A relaxation of strict representational compression

Representations need not always delete input information to be probabilistic. Animals often do not have information *a priori* that task-irrelevant input features for a particular task might not become relevant for a different task, so it would be suboptimal to compress away that information. Therefore, we propose a second metric, called *decoder-specific* compression: instead of requiring performers to delete all irrelevant input information, truly probabilistic representations need only represent posterior information in a way that irrelevant variability can be easily projected to the nullspace of a downstream readout. Then, rather than deleting information, a probabilistic representation could simply cluster features into orthogonal and non-overlapping subspaces, and a fixed decoder would merely require a fixed rotation in order to extract the task-relevant posterior. Formally, we can test this by assessing whether task-relevant posteriors are linearly decodable from neural activity–if they are, there exists a subspace in which only that posterior is represented and nothing more.

## 3 Methods

Our approach requires training two types of neural networks: "performers" and probes. Performers are recognition models trained to perform a particular inference task. Probes are trained on the hidden activations of the performers to evaluate whether their internal representations are probabilistic.

### 3.1 Performer training

Following [9], we trained feedforward "performer" networks to optimally perform cue combination and coordinate transformation; these networks are referred to as *task-optimized performers*. In both tasks, the inputs to the performer network consisted of two neural populations $\mathbf{x}_1, \mathbf{x}_2$, each with 50

independent Poisson neurons that had Gaussian tuning curves. The height of the neural population responses was modulated by a gain $\nu_i$, which was population-dependent and varied trial-by-trial. The activities of the input populations constituted the observations ("cues") based on which the performer networks needed to compute their outputs. In cue combination, both populations were driven by the same latent stimulus $z$, which the network had to estimate based on input layer activities (Figure 2A). In coordinate transformation, each population was driven by a different $z_i$, and the performer had to optimally estimate the sum of the two latent stimuli $z = z_1 + z_2$—a more difficult task considering the network must marginalize out $z_1$ and $z_2$ [27, 28]. The gains $\nu_i$ for each input population were considered nuisance variables akin to psychophysical variables like contrast that the performers needed to marginalize out.

To study probabilistic representations in dynamic inference tasks, we trained *recurrent* performer networks on the simplest form of dynamic state estimation: a 1-D linear dynamical system defined by the equations shown in Figure 2, third panel. Here, $x_t$ and $z_t$ denote the observation and latent state at time $t$, respectively. $Q$ denotes the *process noise variance*, which directly modulates the true state, whereas $\nu$ is the *measurement noise variance*, which has no impact on the true state but makes inference harder. We were interested in evaluating whether recurrent performers trained to perform Kalman filtering would implicitly understand how to weight incoming measurements based on their relative uncertainties. This is especially interesting under resource-constrained conditions where the number of hidden neurons is an order of magnitude smaller than the total number of observations (or total number of time steps $T$) because the network does not have enough capacity to memorize the entire sequence trajectory. For additional details on the generative details for the Kalman filtering experiments, see Appendix.

**Testing generalization**    We evaluated network generalization on unseen nuisance conditions. In the "all nuisances" condition, stationary inference performers were trained and tested on all pairwise nuisance combinations $(\nu_1, \nu_2) \in \mathcal{V} = \{0.25, 0.5, 0.75, 1, 1.25\}$. For the "interpolation" and "extrapolation" conditions, networks were trained on a subset of $\mathcal{V}$ ($\nu_1, \nu_2 \in \{0.25, 1.25\}$ and $\nu_1, \nu_2 \in \{0.25, 0.5\}$, respectively) and tested on the remainder. For Kalman filtering, we used the same train/test splits over $\nu \in \mathcal{V}$.

**Baseline performers**    To compare information content in the internal representations of task-optimized performers, we selected two suitable baseline performers. The first was a copycat network, which trivially copied inputs to its hidden layer. Such a network is sufficient but almost never minimal, making it a natural lower bound on compression. For the static inference tasks, the copycat's input-to-hidden weight matrix was the identity matrix (with zero-padding) and only the hidden-to-output weights were trained. See Appendix for the copycat design in the Kalman filtering task.

### 3.2    Approximating information content via the fIB framework

Classical information bottleneck approaches are critically limited by their reliance on estimating mutual information. To circumvent this, we measure functional information use (rather than mutual information) by training linear and nonlinear posterior (i.e., sufficiency) probes and nonlinear input (i.e., minimality) probes on the hidden activations of fully trained networks (Figure 2B); crucially, while we want posterior probes to perform well, we want input probes to perform poorly in neural representations that are probabilistic (minimal). Posterior probes (nonlinear MLP and linear) were trained to minimize the Kullback–Leibler divergence between predicted and optimal discretized posteriors. For stationary inference tasks, input probes were trained with Poisson negative log-likelihood to reconstruct the full Poisson input population. In Kalman filtering, the input probes were trained with MSE loss on individual input lags (labeled accordingly in Figure 3B). See Appendix for further details about probe training.

## 4    Results

The fIB framework reveals a clear dissociation between probabilistic and heuristic neural representations (Figure 3). Probabilistic population codes (PPC) combine high posterior decodability (ordinate) with strong input compression (abscissa: input decodability), indicating a true posterior representation. In contrast, copycat networks (COPY) also show high posterior decodability but no compression, since they simply pass inputs through to the hidden layer. Task-optimized performers

in stationary inference tasks consistently remain in an input recoding regime: they achieve decent posterior decodability but compress inputs no more than the copycat, even across generalization conditions (columns in Figure 3A). Recurrent performer networks also fail to develop probabilistic internal representations, despite performing state estimation accurately under nuisance generalization. Although they demonstrate some compression deeper in the sequence history (Figure 3B, successive rows), this degree of compression remains well below that of the PPC and follows (qualitatively) the expected exponential memory decay of RNNs. Importantly, posterior decodability is considerably lower than both benchmark performers, and both posterior decodability and input compression *worsen* during training in all generalization conditions. Training checkpoints are color-coded from early (red) to late (blue) learning (in all panels).

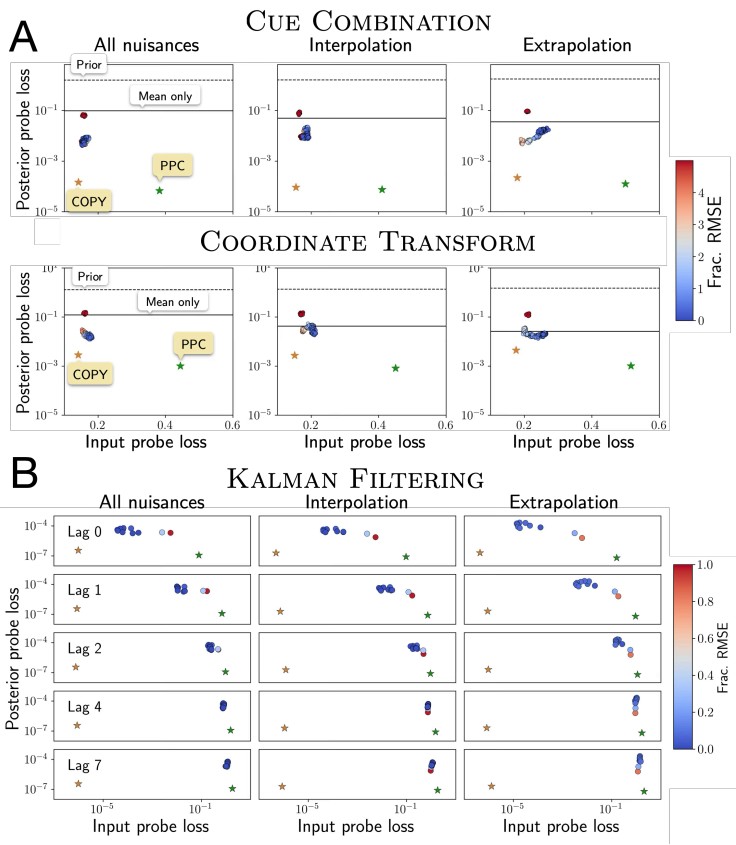

We determined the representational simplicity (linearity) of posterior representations using the relative performance between non-linear and linear posterior probes (supplementary figure 4). In cue combination, linearity is trivial because both a probabilistic population code (PPC) and a copycat network (COPY) are guaranteed to be able to construct the log-posterior linearly if neural variability is in the exponential family of distributions [27]. Accordingly, the network also maintains a linear code throughout learning. More interestingly, in coordinate transformation and Kalman filtering, the copycat network is not able to linearly construct the log-posterior. Here, the task-optimized networks do not seem to converge on a linear representation of the log-posterior, suggesting that learned representations are not invariant to nuisances *even in a linear subspace*.

Figure 3: fIB results for **A)** static inference tasks, and **B)** dynamic inference tasks. Networks do not learn to optimally compress inputs.

These results suggest that neural networks trained with non-probabilistic objectives do not generically develop probabilistic representations. In simple settings, they fail to develop invariant codes that extract minimal sufficient statistics from input observations. We note that applying these methods to neural data remains an open challenge, especially because we cannot always assume the structure of $\mathbf{X}, \mathbf{r}$, or even $\mathbf{y}$ in neural data. Nevertheless, we contend that task-optimal compression is a powerful hallmark of probabilistic computation that can be instrumental in distinguishing probabilistic and non-probabilistic representations in neural networks and uncovering the structure of neural codes of uncertainty. This framework will also be useful in identifying the exact inductive biases that promote probabilistic representation and generalization in artificial networks.

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

# 5 Appendix

## 5.1 Additional performer training details

All task-optimized performers were trained with mean squared error loss and stochastic gradient descent (Adam optimizer) and were trained for 10,001 epochs, which was sufficient for training loss to plateau. Feedforward (static inference) performers contained a single hidden layer with 200 neurons (and ReLU activations) and a final linear readout neuron. Recurrent (dynamic inference) performers consisted of a single gated recurrent unit (GRU) hidden layer, and a final linear readout neuron (since we consider only a 1-D filtering case). The number of hidden neurons in the recurrent layer ($H = 13$) was selected to match the window size of the copycat benchmark network (cf. Section 3.1).

## 5.2 Probe training details

We trained (also with Adam) two types of "probe" networks, posterior probes and input decoders, to assess the structure of the internal representations in the task-optimized performers (Figure 2B). Posterior probes were multilayer ReLU networks (512 hidden neurons) trained (with a KL divergence loss) to decode a discretized version of the ground-truth posterior distribution (which was analytically computable from input layer activations) from the hidden layer activations of the task-optimized networks ($\mathbf{r}_{\text{perf}}$). We chose to decode discretized posteriors—rather than the posterior sufficient statistics (e.g., mean and variance)—because discretization enforces greater structure than simply low-order moments. Although discretization is lossy, it constrained the probe to output valid probability distributions rather than arbitrary numbers (as would be the case if regressing sufficient statistics), which prevented trivial solutions and stabilized training. We then trained a linear posterior probe with the same objective as the nonlinear posterior probe to assess the simplicity/usability of the neural representation. This also assessed whether these networks behaved like probabilistic population codes (á la [27]), which predict that neural activity represents log-posteriors linearly.

Finally, we trained two kinds of input decoders for static versus dynamic inference tasks. For cue combination and coordinate transformation, the input probe was a two-layer ReLU networks with 200 hidden neurons trained to reconstruct the full input layer activations from its hidden layer, and these probes were trained with Poisson negative log-likelihood loss.

Given the temporal nature of dynamic inference tasks, decoding the entire input sequence—*i.e.,* a dynamical system trajectory—at each time step would not be appropriate. Instead, we trained a set of probes, each tasked with decoding a *fixed* lag in the sequence. For example, the lag 0 probe was, at every time step in the sequence, trained to predict the most recent measurement. We used lags $l = 0, 1, 2, 4$, and 7, which meant that input probe training began only after a burn-in of 7 time steps in each trial; without this, comparing the lag 0 and lag 7 probes would be unfair, as the former would see more measurements. As these probes were each trying to decode only the scalar raw measurement at lag $l$, they were trained using mean squared error loss on the true measurement at lag $l$.

The training objectives for each for the probes is as follows:

**Posterior probe**

$$\underset{\mathbf{W}_{\text{post}}, \mathbf{w}_{\text{post-readout}}}{\arg\min} \quad D_{\text{KL}} \left( p_{\text{true}}(z|\mathbf{x}_1, \mathbf{x}_2) \| \underbrace{\text{Softmax}(\mathbf{w}_{\text{post-readout}}[\mathbf{W}_{\text{post}}\mathbf{r}_{\text{perf}}]_+)}_{\tilde{p}(z|\mathbf{x}_1, \mathbf{x}_2)} \right)$$

**Linear posterior probe**

$$\underset{\mathbf{w}_{\text{lin}}}{\arg\min} \quad D_{\text{KL}} \left( p_{\text{true}}(z|\mathbf{x}_1, \mathbf{x}_2) \| \underbrace{\text{Softmax}(\mathbf{w}_{\text{lin}}\mathbf{r}_{\text{perf}})}_{\tilde{p}(z|\mathbf{x}_1, \mathbf{x}_2)} \right)$$

**Input probe**

$$\underset{\mathbf{W}_{\text{input}}, \mathbf{w}_{\text{input-readout}}}{\arg\min} \quad \mathcal{L}_{\text{NLL}}\left((\mathbf{x}_1\,\mathbf{x}_2), \underbrace{\text{Softplus}(\mathbf{w}_{\text{input-readout}}[\mathbf{W}_{\text{input}}\mathbf{r}_{\text{perf}}]_+)}_{=(\hat{\mathbf{x}}_1\,\hat{\mathbf{x}}_2)}\right)$$

For Kalman filtering, the nonlinear posterior and input probes had an additional ReLU layer (three in total). However, the nonlinear (and linear) posterior probe otherwise shared the same structure as those used for the static inference tasks. The input probe, however, was only trained to decode particular lags. Therefore, one input probe was trained for each lag shown in Figure 3B. The input probes were trained with mean squared error loss on the true measurement of the appropriate lag $l$.

**Input (lag) probe**

$$\underset{\mathbf{W}_{\text{input}}, \mathbf{W}_{\text{hid-input}}, \mathbf{w}_{\text{input-readout}}}{\arg\min} \quad \mathcal{L}_{\text{MSE}}\left(x_{t-l}, \underbrace{\mathbf{w}_{\text{input-readout}}[\mathbf{W}_{\text{hid-input}}[\mathbf{W}_{\text{input}}\mathbf{r}_{\text{perf}}]_+]_+}_{=\hat{x}_{t-l}}\right)$$

## 5.3 Constructing the PPC benchmark

We have a log-posterior $\rho_z = \log p(z|\mathbf{X})$. PPC literature suggests that log-posteriors are a linear function of neural activity, *i.e.,*

$$\mathbf{A}\mathbf{r}_{\text{PPC}} + \mathbf{b} = \rho_z,$$

where $\mathbf{r}_{\text{PPC}}$ represents the hidden activations of the PPC, $\mathbf{A}$ is a matrix of tuning curves, and $\mathbf{b}$ is a bias term. Thus, to construct the PPC hidden layer, we wanted to find the hidden activations $\mathbf{r}$ corresponding to $\rho_z$. This is a simple least squares optimization problem that yields the well-known Moore-Penrose pseudoinverse:

$$\mathbf{r}_{\text{PPC}} \approx (\mathbf{A}^\top\mathbf{A})^{-1}\mathbf{A}^\top(\rho_z - \mathbf{b}).$$

For a well-conditioned and appropriately chosen (but otherwise generic) $\mathbf{A}$, this provides a valid representation of the optimal log-posterior. In practice, we assume that $\mathbf{A}$ consists of Gaussian tuning curves evaluated at a (sufficiently large) discrete set of latent stimulus values. This choice is natural because the posteriors in all tasks studied herein are Gaussian (or approximately so), and Gaussian tuning curves provide a robust basis for representing them. For a fair comparison across experiments, $\mathbf{r}_{\text{PPC}}$ was always constructed to match the dimensionality of the hidden layer of the task-optimized performer.

## 5.4 Designing a copycat network for Kalman filtering

In the static inference tasks, our copycat directly represented entire input patterns in its hidden activations. Extending this idea to Kalman filtering required additional care because of the temporal structure of the inputs. If the copycat were to present the entire input sequence to its hidden layer at every time step, then the hidden activity would remain identical across time within a trial, making the probing analysis ill-posed. Another alternative is for the copycat to represent only the observations revealed up to time $t$, so that hidden activity evolves with time. However, this raises the complication of appropriately handling the yet-unobserved future measurements—while zero-padding may be a natural solution, zero is itself a valid measurement value, so it conflates "unobserved" with "observed = 0," potentially leaking spurious information into the hidden representation. Hence, we instead designed the copycat network as a sliding window of size $w$ that was an order of magnitude smaller than the total number of time steps $T$. Window size was chosen by finding the minimum $w$ for which a sliding-window Kalman filter—a Kalman filter with limited memory horizon—of length $w$ approximated the full Kalman filter up to a mean squared error cutoff of $10^{-9}$ (Supplementary figure 6). Then, during training, the first $w$ time steps of each trial were burnt-in before training for the fIB probes commenced.

## 5.5 Computing the 'mean only,' fixed posterior variance performer benchmark

We want to compare the information content of our performers to a non-probabilistic heuristic model that assumes *fixed* trial-to-trial posterior variance but is able to perfectly decode the posterior mean (a 'mean-oracle', shown in Figure 3A as the horizontal dashed lines). This derivation assumes that the posteriors are Gaussian (or can be well-approximated with Gaussians). This is an appropriate assumption for our tasks, given that for the static inference tasks, we use Gaussian tuning curves and Poisson neural variability, which yield approximately Gaussian posteriors [27].

Assume that the true posterior on trial $i$ is $p_i(z) = \mathcal{N}(z|\mu_i, \sigma_i^2)$, and we want to approximate this with our fixed-width mean oracle $q_i(z) = \mathcal{N}(z|\mu_i, \hat{\sigma}^2)$. What should $\hat{\sigma}^2$ be?

$$\langle D_{KL}(p_i||q_i)\rangle_i = \frac{1}{2N}\left[\sum_i\left(\frac{\hat{\sigma}^2}{\sigma_i^2} + \ln\frac{\sigma_i^2}{\hat{\sigma}^2}\right) - N\right]$$

$$\frac{\partial}{\partial\hat{\sigma}^2}\langle D_{KL}(p_i||q_i)\rangle_i = \frac{1}{2N}\left(\sum_i\frac{1}{\sigma_i^2} - \frac{1}{\hat{\sigma}^2}\right)$$

$$0 = \sum_i\frac{1}{\sigma_i^2} - \frac{N}{\hat{\sigma}^2}$$

$$\frac{N}{\hat{\sigma}^2} = \sum_i\frac{1}{\sigma_i^2}$$

$$\hat{\sigma}^2 = \frac{N}{\sum_i\frac{1}{\sigma_i^2}}$$

## 5.6 Additional Kalman filtering details

The results presented in this paper fix $A = 0.75, H = 1.0, Q = 0.5$, and $T = 40$. To mimic the nuisance "gains" from the PPC-style static inference tasks, we modulated measurement noise $\nu \in \{0.25, 0.5, 0.75, 1.0, 1.25\}$ from trial-to-trial (and fixed it within a trial). Because the performers were not given explicit information about $\nu$ from trial-to-trial, we consider the *marginal* Kalman filtering posterior whenever we performed our fIB analysis—that is, the posterior $p(z_t|x_{1:t}, \nu)$ marginalized over $\nu$. Therefore, posterior uncertainty reflects not just uncertainty about the state estimate, which in Kalman filtering is monotonic and independent of the measurements $x_{1:t}$, but also uncertainty in $\nu$ itself, which makes the optimal posterior variance a function of the measurements and, thus, causes it to be temporally non-monotonic.

### 5.6.1 Deriving the marginal Kalman filtering posterior

Here, we take the standard Kalman filter, which is, importantly, conditioned on the parameters $\theta$ of the linear dynamical system, and marginalize out $\theta$. In our case, $\theta = \nu$, but this approach can be applied to any of the parameters in $\boldsymbol{\theta} = \{A, H, Q, \nu\}$.

$$p(z_{t+1}|x_{1:t+1},\theta) = \int p(z_{t+1}, z_t|x_{t+1}, x_{1:t}, \theta)dz_t$$

$$\propto \int p(x_{t+1}|z_{t+1}, z_t, x_{1:t}, \theta)p(z_{t+1}|z_t, x_{1:t}, \theta)p(z_t|x_{1:t}, \theta)dz_t$$

$$= \int p(x_{t+1}|z_{t+1}, \theta)p(z_{t+1}|z_t, \theta)p(z_t|x_{1:t})dz_t$$

$$= p(x_{t+1}|z_{t+1}, \theta)\int p(z_{t+1}|z_t, \theta)p(z_t|x_{1:t})dz_t$$

$$= \pi(z_{t+1}|x_{1:t+1}, \theta)$$

$$p(x_{t+1}|x_{1:t}, \theta) = \int p(x_{t+1}, z_{t+1}|x_{1:t}, \theta)dz_{t+1}$$

$$= \int p(z_{t+1}|x_{1:t+1}, \theta)dz_{t+1}$$

$$= \bar{\pi}(x_{1:t+1}, \theta)$$

$$p(z_{t+1}|x_{1:t+1}, \theta) = \frac{\pi(z_{t+1}|x_{1:t+1}, \theta)}{\int \pi(z_{t+1} = z'|x_{1:t+1}, \theta)dz'}$$

$$= \frac{\pi(z_{t+1}|x_{1:t+1}, \theta)}{\bar{\pi}(x_{1:t+1}, \theta)}$$

$$p(x_{1:t+1}|\theta) = p(x_1, ..., x_t, x_{t+1}|\theta)$$

$$= p(x_1|\theta)p(x_2|x_1, \theta)p(x_3|x_2, x_1, \theta)...$$

$$= \prod_{\tau=1}^{t} p(x_{\tau+1}|x_{1:\tau}, \theta)$$

$$= \prod_{\tau=1}^{t} \bar{\pi}(x_{1:\tau+1}, \theta)$$

$$p(\theta|x_{1:t+1}) = \frac{p(x_{1:t+1}|\theta)p(\theta)}{\int p(x_{1:t+1}|\theta)p(\theta)d\theta}$$

$$= \frac{p(\theta)\prod_{\tau=1}^{t} \bar{\pi}(x_{1:\tau+1}, \theta)}{\int \prod_{\tau=1}^{t} \bar{\pi}(x_{1:\tau+1}, \theta)p(\theta)d\theta}$$

$$p(z_{t+1}|x_{1:t+1}) = \int p(z_{t+1}|x_{1:t+1}, \theta)p(\theta|x_{1:t+1})d\theta$$

## 5.7 Supplementary figures

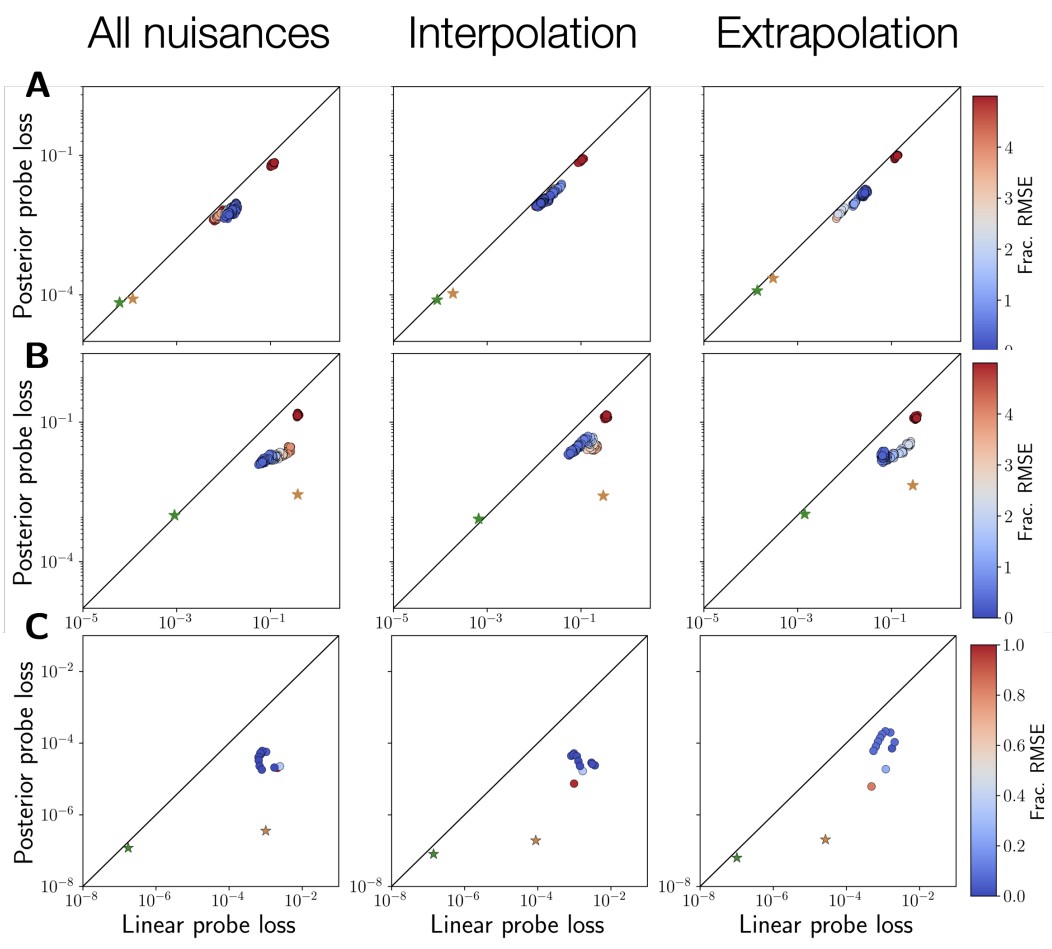

Figure 4: **Task-optimized performers do not form strongly linear representations in general.** Posterior and linear posterior probe loss are plotted against each other for **A.** cue combination, **B.** coordinate transformation, and **C.** Kalman filtering. Scatter plot and benchmark performer colors are the same as in Figure 3. The solid line indicates unity of linear and nonlinear posterior probe performance, signifying a simple linear internal representation.

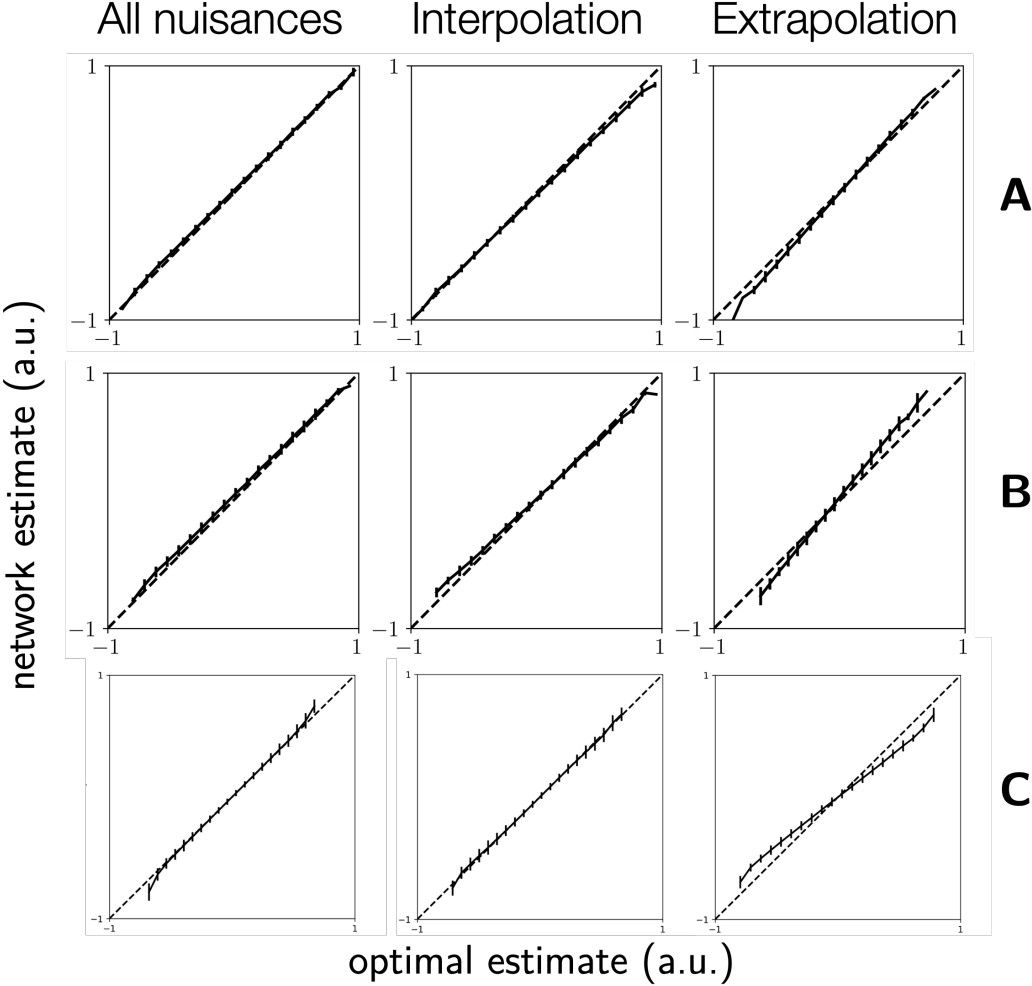

Figure 5: **Performers consistently behave in a Bayes-like manner, even under out-of-distribution nuisance generalization.** Performers output Bayes-optimal inferences in **A.** cue combination, **B.**, coordinate transformation, *and C.* Kalman filtering. For the two nuisance generalization conditions tested in [9] ("all nuisances" and "interpolation"), performers are robustly Bayesian. Under a third generalization condition, "extrapolation," performance degrades mildly across all three tasks but still remains qualitatively similar to the Bayes-optimal estimates.

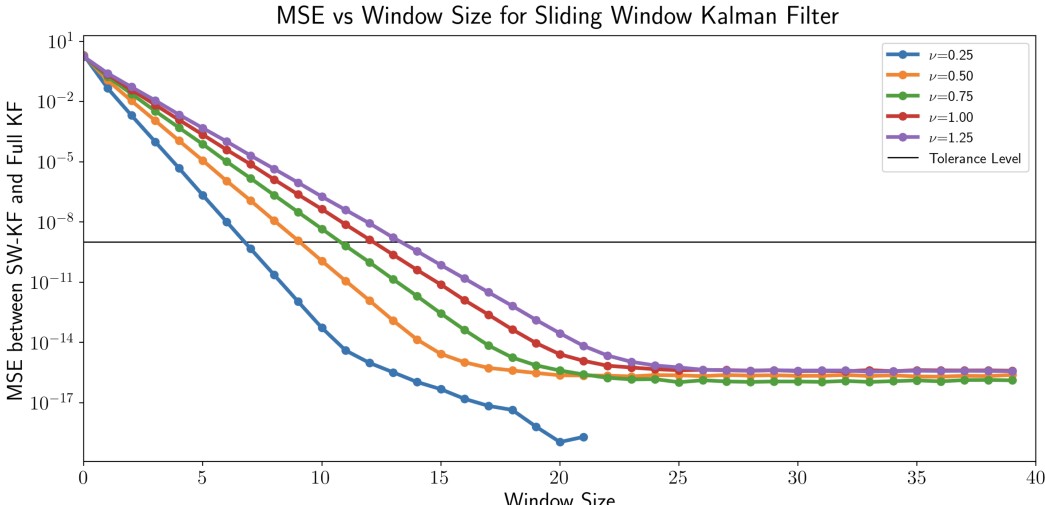

Figure 6: **Number of hidden neurons in the recurrent performer was chosen by comparing a sliding-window Kalman filter to a full Kalman filter.** For each nuisance parameter value $\nu$, we compared how close a sliding-window Kalman filter was to the full Kalman filter as a function of the window size for the sliding-window filter. Using a cutoff of $10^{-9}$, we selected a window size (or number of hidden recurrent neurons) of 13 for all performers.

