# OpenReview forum: "Distinguishing probabilistic from non-probabilistic neural representations"
_NeurIPS.cc/2025/Workshop/UniReps — UniReps2025_

### Official Review · Reviewer_MQk5 · 2025-09-09
**Review of "Distinguishing probabilistic from non-probabilistic neural representations"**

**Confidence:** 4

**Review:**

This extended abstract reexamines the results of a 2017 study by Orhan and Ma, which claimed that neural networks trained without explicitly probabilistic feedback developed probabilistic representations of task uncertainty. The main claim of the work is that trained networks do not in general learn "truly probabilistic" representations, as they retain in their representations irrelevant information about their inputs, and in some cases seem to fail to encode the width of the posterior. I think this work is interesting, and of clear relevance to UniReps, but I have some concerns regarding the methodology and the interpretation of the results.

**Major comments**

- I have a basic qualm about the definition of "truly probabilistic representation" adopted by the authors, an issue which they themselves indirectly acknowledge in Lines 156-160. When considering a hidden representation of a neural network that is larger than the subsequent layer, one must take into account the substantial flexibility this hidden layer has to preserve information in a way that is entirely irrelevant for the function of the network by shunting it into the null space of the weight matrix of the next layer. Thus, it is entirely plausible—and indeed consistent with the authors' findings—that a non-invariant representation may appear to the downstream layers as "truly probabilistic" because task-irrelevant information is contained in the null space. However, given access to the full hidden layer, this irrelevant information is easily recovered. From an optimization perspective, there seems to be little incentive for the network to compress away this null-space information, as by definition it doesn't affect the output. In my mind, this weakens the authors' argument. I think more prose and more analyses are required to address this point:
    - First, the authors should discuss early on (in section 2) this issue of functionally-irrelevant variance.

    - Second, it is important to perform a more direct test of whether the output-potent component of the representation is relatively invariant. For a linear decoder, this could be accomplished by computing the projections of the representation into output-potent and output-null subspaces, and then separately training decoders from each.

- The key methodological choice of this work is to assess invariances through the performance of decoders trained to minimize an estimate of the KL divergence between their output and a discretized version of the posterior distribution, or of the input distribution. This choice requires more conceptual justification, and also perhaps more control experiments. The authors do not justify why they use a discretization-based estimator (indeed, this to some degree goes against the discussions in Lines 93-95, given the issues with binning discussed in the debate around applications of IB to neural networks, to which I'd add [Saxe et al. (ICLR 2018)](https://openreview.net/forum?id=ry_WPG-A-)). Using something like a Wasserstein distance or an $f$-divergence (see e.g. [Tschannen et al. (ICLR 2020)](https://openreview.net/forum?id=rkxoh24FPH)) would almost certainly be more robust, and would avoid discretization issues.

- The figures—particularly Figure 2—are too small as to be easily legible. Reducing figure size to this degree is not an acceptable approach to meeting length restrictions. Moreover, the font sizes in the two halves of Figure 2 differ, which should be rectified.

**Minor comments**

- Are the ordinates of all panels under "Representational Simplicity" in Figure 2 the posterior probe loss? It might be helpful to clarify this.

- In Lines 99 and 112, why do you refer to “performer” networks? This risks confusion with the Performer architecture introduced by [Choromanski et al (ICLR 2021)](https://arxiv.org/abs/2009.14794); indeed this confused me on first reading.

- In line 43, the notation $\mathbf{y} = p(y\mid\mathbf{X})$ seems to mix random variables and probability densities. I presume you mean to define $\mathbf{y}$ as the posterior estimate of $y$ given observations $\mathbf{X}$?

**Score:**

2

**Topic Fit:**

3

---

### Official Review · Reviewer_rEan · 2025-09-12
**Original and potentially important work; more more can be done to make it clear and accessible to a broad audience**

**Confidence:** 3

**Review:**

The authors test whether task-optimized neural networks spontaneously develop probabilistic representations, as suggested by previous work. The exact definition of what constitutes probabilistic vs. non-probabilistic representation is notoriously difficult to pinpoint. This paper adopts a reasonable approximation based on the concepts of invariance and representational simplicity (probabilistic representations should be high on both). They show that a network that computes a Probabilistic Population Code has these properties, whereas a network that simply represent the input (Copycat network) has low invariance. Based on these criteria, the authors demonstrate that networks trained to perform standard tasks (cue combination and coordinate transformation) don't develop probabilistic representations.

This is an important result, especially since it provides a counterweight to previous claims. It doesn't solve the problem of how exactly to ensure that a representation in a neural network is or isn't probabilistic, but it does provide some useful heuristic rules in making such judgments. This work is thus important for the debate regarding whether human perception is probabilistic.

The work is highly original, but more can be done to make it clear and accessible to a broad audience.

**Score:**

4

**Topic Fit:**

3

---

### Official Review · Reviewer_QjJS · 2025-09-15

**Confidence:** 2

**Review:**

# Summary

The paper proposes an IB-inspired minimal sufficiency criterion for labeling representations as probabilistic:
a representation must be sufficient for the target posterior and minimal in input information, which entails invariance to nuisances.
This sharpens prior “emergent probabilistic representation” claims that leaned on decodability (you can read out a posterior) alone.
Empirically, the authors train standard MLPs on cue-combination and coordinate-transform tasks with nuisance gain, then probe hidden states with (i) a posterior probe and (ii) an input decoder.
They introduce two baselines: (a) a hand-coded baseline that shows high decodability with compression, and (b) a second baseline that shows decodability but bypasses input-encoding (pass-through/no compression).
They then showed that typical task-optimized performers follow the second baseline: they have high decodability without compression, i.e., input re-representation rather than probabilistic coding; training often converges to a mean-oracle solution.
Thus, decodability is necessary but not sufficient; compression/minimality is showcased to be the key additional test to substantiate probabilistic representations.


# Strengths
- **Novelty**:
The paper’s strengths lie in offering a clear, operational criterion for probabilistic internal codes:
representaions should both (i) make the target posterior easily readable and (ii) drop unnecessary input detail (an information bottleneck inspired minimality view).
Furthermore, the authors couple this with a simple, practical two-probe protocol (posterior probe and input reconstructor) that avoids mutual information estimation and works for deterministic networks.

- **Empirical evaluation**:
The two baselines bracket behavior well:
  1. the positive PPC baseline showing decodability with compression and
  2. the negative copycat baseline showing decodability without compression.

  Additionally, the nuisance gain setups (all gains, interpolation, and extrapolation) verify whether representations actually discard nuisance information, rather than allowing the readout to compensate.

- **Potential to inspire discussion**:
Finally, the finding that task-optimized networks tend to re-represent inputs instead of compressing is a clear and helpful negative result.
While it remains open which inductive biases produce probabilistic representations, the proposed test provides a concrete way to gather evidence that a given bias promotes compressed probabilistic codes.


# Weaknesses
- **Clarity**:
My main concern is clarity and accessibility:
several key terms (e.g., “probabilistic representation”, “inference as compression”, “code”, “two-cue fusion”) are used in ways that differ from common ML usage or just underexplained.
Also, the initial connection from the neuroscience motivation to the ML probing setup feels weak.
A short glossary and a simple ML problem statement would help.

- **Reproducibility**:
Methodologically, some details are sparse:
How exactly is the ground-truth posterior computed and discretized for the KL probe?
How is the PPC baseline constructed? Just missing training and experimental details in general.

- **Complexity of the experiments**:
The empirical scope mirrors prior “emergence” setups (vanilla feed-forward MLPs on two synthetic tasks) to enable a direct comparison. This is appropriate for the paper’s aim of proposing a test, but it means we don’t yet know how the conclusions change under stronger inductive biases; applying the test in those settings is promising future work.

None of these issues feels critical, especially for a 4-page extended abstract. Still, a mini-glossary, one or two clarifying sentences, an explicit posterior formula or code link, and a brief robustness note would greatly improve readability and confidence.
My confidence in these critiques is low due to my limited background, but the suggested additions would make the work easier to evaluate across subfields.

# Recommendation
**Weak Accept.**
The paper provides a useful correction to how the community tests for “probabilistic” representations, supported by clean baselines and a practical probe protocol.
Clarity and reproducibility are my main concerns, which probably hinder access for a broad NeurIPS audience, but these can be addressed with minor adjustments.
Considering the Extended Abstract track and its non-archival nature, the conceptual contribution and empirical results are strong enough to warrant acceptance.

**Score:**

3

**Topic Fit:**

3

---

### Official Review · Reviewer_gu3f · 2025-09-16
**Well-framed problem, elegant and simple approach to probe it**

**Confidence:** 4

**Review:**

**Quality:** Solid experimental framing (performer vs. copycat vs. PPC) and a practical probe toolkit when mutual information is ill-posed in deterministic nets. Conclusions rest on probe behavior, results would be stronger with systematic sweeps on probe capacity/regularization and clearer uncertainty reporting.

**Clarity:** Motivation and setup are well explained. The main claims are easy to follow. Figure 2 is not very readable.

**Originality:** The conceptual move is to elevate minimality as a necessary criterion and to use the copycat baseline to show why decodability alone is insufficient. The concrete counterexample is helpful.

**Significance:** Offers a useful standard that can change how people assess “probabilistic” representations, especially in settings where posterior readout looks good but invariance/compression is unclear.

---

**Pros**
- Clear statement of the problem: sufficiency isn’t enough, minimality matters for robustness and transfer.
- Simple probing recipe (posterior probes vs. input-reconstruction decoders) that others can replicate.
- Learning-trajectory analyses are nice (posterior decodability can improve before compression).

**Cons**
- It seems like you are using a different definition for what probabilistic representation is, which although you have intuitive explanations, I am not convinced to not call the other mentioned papers as probabilistic representations, because it all boils down to how you define it. In this sense, I found the title a little misleading.
- Narrow experimental scope (simple tasks, shallow architectures); unclear how this carries to modern models.
- Some operational details are brief, which makes reproducibility tougher.

**Score:**

4

**Topic Fit:**

3